# Environmental Impacts of Egg Production from a Life Cycle Perspective

Aurore Guillaume [1,2,*], Anna Hubatová-Vacková [1] and Vladimír Kočí [1]

1 Department of Product Ecology and Sustainability, University of Chemistry and Technology, Technická 5, 166 28 Prague, Czech Republic; hubatova@vscht.cz (A.H.-V.); kociv@vscht.cz (V.K.)
2 Process and Environmental Technology Lab, Department of Chemical Engineering, KU Leuven, Jan Pieter De Nayerlaan 5, B-2860 Sint-Katelijne-Waver, Belgium
* Correspondence: guillaua@vscht.cz

**Abstract:** The food industry represents a vast environmental burden to our planet. Most animal products are known to have greater environmental impacts than alternative plant-based sources of nutrition. One of the most consumed animal products are eggs, represented in most dietary habits both as a primary ingredient and processed. In the European Union (EU), eggs for consumption can be produced in four different laying systems: enriched cages, barns, free-range, and organic. In past years, discussions about the ethical perspective have been ongoing among the wider public, industry and academia. However, the essential comparison of environmental impacts of the laying systems has been missing in our region. Life Cycle Assessment (LCA) is an excellent tool for comparing environmental impacts of various scenarios. Therefore, we performed a LCA of several egg-producing subjects in the Czech Republic, representative of all four laying systems. In addition, these regulated laying systems were compared to a community garden system. Our results suggest feed conversion ratio (FCR), feed composition, and manure management to be the most important factors influencing the total environmental impacts of eggs. Moreover, environmental benefits linked to outdoor access or using organic feed over conventional were observed in our study.

**Keywords:** Life Cycle Assessment; egg; laying hen systems; environmental impacts; poultry; food production

## 1. Introduction

Eggs are one of the most consumed animal products and represent an important source of high-quality protein and micronutrients in the human diet. They are considered healthy, nutritious, and are also popular due to their low price [1]. In total, approximately seven million tons of eggs are produced annually in the EU [2]. Behind such a widely consumed product stands a big industry with significant environmental impacts.

It has been shown by Poore and Nemecek [3] that a diet containing animal products has greater environmental impacts than a plant-based one. At the same time, eggs represent a suitable alternative to replace less healthy animal products with even more extensive environmental impacts, such as processed red meat [4]. Therefore, the rising popularity of vegetarian diets could result in a future increase in egg consumption [2]. Ethical issues linked to egg production are commonly discussed, while its environmental concerns have been slightly neglected.

Various studies assessing egg-laying systems from the environmental perspective have been performed. In the EU, eggs are produced in four different laying systems: enriched cage, barn, free-range, and organic. Some studies evaluated the environmental impacts of a single laying system, such as intensive egg production in Spain [5], organic eggs in Italy [6], some farms in Iran [7] or a comparison between caged and free-range eggs in Australia [8]. A comparison between the laying systems has been carried out in the

Netherlands [9] and the United Kingdom [10]. However, most data in the first study [9] were taken from databases or publications, while the second study [10] was carried out before the European laws banned conventional cages. The only recent study comparing the environmental impacts of laying systems was performed in Canada [11], where farming regulations differ significantly.

LCA is a widely used tool for assessing environmental impacts of products, services, or organizations, including food products [12]. Here, it has been used to evaluate the environmental impacts of nine model farm subjects in the Czech Republic, representative of all four egg laying systems. In addition to that, one community garden farm was assessed. To improve sustainability of egg production, mitigation potential of replacement of key parameters such as feed and yield were estimated.

The functional unit was defined as one kg of shelled eggs. The system boundaries of this study were from cradle-to-gate, i.e., the life cycle of eggs from hatching until the end of the agricultural phase. The inputs included e.g., feed production, thermal energy and electricity production, water consumption, or transportation between stages. Waste and manure disposal or wastewater treatment were among the assessed outputs. The packaging, use of drugs, distribution, and consumption phases were not considered.

We believe that this study performed in the Czech Republic is of interest for the entire EU region, its authorities, and policymakers, as well as for the general public. As we are nearing the EU ban of caged laying systems, it may help battery farms in considering their transformation into another laying system. While the obtained data can be used in creating strategies for sustainable development, an increased awareness about the environmental impacts of egg laying systems is also essential for conscious consumption choices.

## 2. Materials and Methods

### 2.1. Goal and Scope of the Study

2.1.1. Objective and Functional Unit

The objective of this study was to assess environmental impacts of egg production in the Czech Republic from cradle-to-gate. In particular, the study aimed to compare the different laying hen systems: battery, barn, free-range, and organic. The functional unit is one kilogram of shelled eggs.

2.1.2. System Description

Firstly, during breeding, hens are selected mainly according to their high rate of egg production, and their eggs are incubated in a hatchery. After hatching, these young chicks are transported to a rearing house, where they will be fed until maturity. EU regulations state that organic laying hens should be provided by organic rearing [13]. However, due the fact that organic pullets are not available in sufficient quantity, it was possible to use non-organic pullets until the end of 2021 [13]. At around 17 weeks of age, hens are sent to a laying house, where they start their productive cycle. There are four types of laying house: battery, barn, free-range, and organic. These differ in the amount of space the hens have, as well as their accessibility to the outdoors or feed. To differentiate between the laying systems, a code number from zero to three is marked on each egg.

In the Czech Republic, battery is the most common type of laying system, representing about 67% of the total laying systems. Barn systems constitute about 31%, while free-range and organic systems constitute only 1% and 0.4%, respectively [2].

In this study, three battery, two barn, and two free-range farms were assessed. There are industrial farms with up to 600,000 laying hens. For organic eggs, two small farms were evaluated. The main activity of the first one is to produce cheese from goats and sheep, but also to grow fruits and vegetables. They feed their hens with wheat and potatoes from their land, as well as leftovers from their bed and breakfast and whey. The second organic farm produces meat and cereals together with eggs. To feed their laying hens, they give a mixture: half cereals from their farm, and half a processed grain mix.

In addition to the four laying systems mentioned, egg production in a community garden (laid in a free-range system) was assessed. The community garden is located in Prague, and there are around 130 hens. Besides the pullets bought, half of their hens are exhausted hens saved from industries, and some other were born on the farm. They use water from the river. Detailed data for each of the farms is given in the Section 2.2.4.

### 2.1.3. System Boundaries

LCA was carried out from the hatchery till the laying of eggs. The breeding step was excluded due to a lack of data, but as it is similar for all systems, it does not affect the comparison. Packaging, distribution, and consumption phases were not considered, and nor were the maintenance of buildings and machines. Information on the use of antibiotics was missing, therefore it was not included in the study. The exhausted hens going to slaughterhouse were considered, unlike the ones sent to a rendering plant. A description of the system boundaries can be found in Scheme 1.

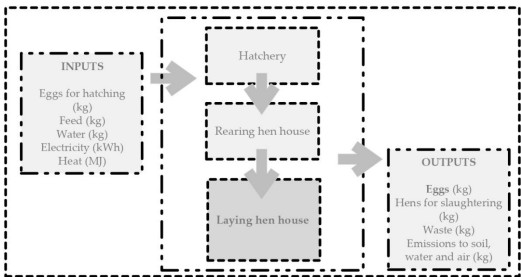

**Scheme 1.** System boundaries for the cradle-to-gate LCA of Czech egg production.

### 2.1.4. Allocation

The environmental impact was completely restricted to eggs. Mass and economic allocations were performed, and it was concluded that the impact of meat from exhausted laying hens is not significant compared to eggs. The mass allocation factor would allocate approximately 5% (for the battery system) up to 10% (for the organic system) of impacts to the meat. For the economic allocation, the price for hens is 5.472 CZK per one kilogram of live weight, and for eggs 1310 CZK for 1000 pieces [14]. An economic allocation would reduce impacts allocated to eggs from approximately 1% for battery up to 2.5% for organic. Furthermore, as it is discussed in Constantini et al. [6], eggs are the main product of the system, and the most important in terms of nutrition, mass and economic values. Meat from exhausted laying hens is considered a low-quality product.

### 2.2. Life Cycle Inventory

### 2.2.1. Data Sources

Hatchery and rearing (in battery and barn systems), as well as battery, barn, free-range laying hen house data, and feed composition for these three laying systems were taken from the Czech Environmental Information Agency (CENIA) portal, which indexes farms emitting more than five tons of ammonia annually [15].

Although large-scale organic laying farms exist in the Czech Republic, most of them belong to a single owner who was not willing to provide any kind of data. However, small organic farms are still representative of the Czech laying systems and data for organic eggs were collected directly from the farms, just as in the case of the community garden.

Processes for energy, feed, wastewater treatment, or other background processes were derived from ecoinvent 3.7 [16] and the GaBi database [17]. Data for the slaughterhouse process were taken from Skunca et al. [18].

### 2.2.2. Assumptions

Hatchery is the same for all laying systems. For this study, two rearing facilities were modelled, one in a battery system with hens intended for battery and barn eggs, and one in

a barn system with hens intended for free-range and organic eggs. The rearing of pullets intended for organic eggs was modelled with 80% of organic feed. For the community garden, hatchery and rearing in a barn system were included for the bought hens, while only rearing was included for hens born in the farm. None of the above was accounted for the exhausted hens, as impacts are allocated to previous egg farms.

We assumed feed was the same for battery, barn, and free-range systems. The ingredients representing less than 1% of the total feed composition were excluded. As mentioned in CENIA documents, it was considered that there is a loss of 4% of hens sent to rendering plants and thus, 96% of hens are sent to slaughter at the end of their production cycle.

For the transport, 50 km between each process was assumed, and the transportation of feed was not considered.

For free-range and organic farms, half of the manure was assumed to be excreted outside and not subject to manure management. For all systems, the manure was managed in a dry system and intended to be applied.

### 2.2.3. Calculations of Emissions

To calculate nitrogen excretion, a mass balance of nitrogen in feed intake was performed. The value of dry matter, crude protein, and phosphorus in feed ingredients were taken from Feedipedia [19]. Following IPCC guidelines [20], we assumed that the fraction of nitrogen intake retained in the body by laying hens was 0.3. The nitrogen applied on soils was calculated by subtracting nitrogen emitted during manure management from nitrogen excreted.

To calculate the amount of phosphorus excreted, a regression was used, which can be found on the Best Available Techniques reference document for intensive poultry rearing [21].

Emissions of ammonia, nitrous oxide (direct and indirect), nitrate, and methane during manure management, application, and deposition on grassland were calculated using Tier 2 equations from IPCC guidelines [20]. The emission factor for methane in dry management system is 0.3 kg $CH_4$/head/year. For leaching of nitrate during manure storage, a country specific value was taken from the National Greenhouse Gas Inventory Report from the Czech Republic [22], which suggests a leaching of 1% of nitrogen excreted. Following the Agrifootprint report [23], 5% of phosphorus was considered as running off to soil.

### 2.2.4. Data Inventory

Table 1 shows data used for the hatchery and rearing houses. Inputs and outputs for the different laying systems can be found in Table 2. The feed composition for rearing and laying systems is described in Table 3.

**Table 1.** Annual inventory data for the hatchery and rearing houses.

| | Hatchery | Rearing (Battery) | Rearing (Barn) | Rearing (Community Garden) |
|---|---|---|---|---|
| | INPUT | | | |
| Area ($m^2$) | 10,412 | 1147.9 | 1256.8 | 800 |
| Eggs (million pieces) | 1200.5 | - | - | - |
| Pullet (number) | - | 29,200 | 25,920 | 15 |
| Electricity (KWh) | 7200 | 167,100 | 182,900 | 0 |
| Heat from natural gas (GJ) | 21,682 | 72.6 | 79.4 | 0 |
| Water ($m^3$) | 21,600 | 3072.6 | 2727.4 | 1.278 |
| Feed (t) | | 397.3 | 352.7 | 0.876 |
| | OUTPUT | | | |
| Pullet (number) | 120,000,000 | - | - | - |
| Wastewater ($m^3$) | 21,600 | 19.1 | 16.9 | - |
| $N_2O$ (kg/hen) | - | 0.0051 | 0.0051 | 0.0052 |
| $NH_3$ (kg/hen) | - | 0.151 | 0.151 | 0.127 |
| $NO_3^-$ (kg/hen) | - | 0.197 | 0.197 | 0.162 |
| $CH_4$ (kg/hen) | - | 0.030 | 0.030 | 0.015 |
| $P_2O_5$ (kg/hen) | - | 0.0027 | 0.0027 | 0 |

**Table 2.** Annual inventory of the farms.

| | Battery 1 | Battery 2 | Battery 3 | Barn 1 | Barn 2 | Free Range 1 | Free Range 2 | Organic 1 | Organic 2 | Community Garden |
|---|---|---|---|---|---|---|---|---|---|---|
| | | | | | INPUT | | | | | |
| Area inside ($m^2$) | 1134 | 2597 | 7546 | 79,010 | 11,776 | 9900 | 4344 | 6 | 25 | 14 |
| Area outside ($m^2$) | 0 | 0 | 0 | 0 | 0 | 600,000 | 224,000 | 150 | 2500 | 786 |
| Hens (number) | 32,000 | 102,400 | 300,000 | 600,000 | 598,732 | 150,000 | 56,000 | 40 | 100 | 130 |
| Electricity (KWh) | 59,839 | 140,921 | 655,000 | 1,455,330 | 1,455,330 | 367,500 | 137,715 | 0 | 0 | 0 |
| Feed (t) | 1600 | 4298 | 12,384 | 25,800 | 25,700 | 6077 | 2351 | 2.34 | 5.475 | 7.59 |
| Water ($m^3$) | 1600 | 8532 | 24,875 | 66,729 | 51,187 | 15,570 | 5803 | 3.358 | 12.775 | 11.072 |
| | | | | | OUTPUT | | | | | |
| Eggs (number/hen) | 352 | 328.5 | 307 | 287.54 | 300 | 299 | 325 | 228 | 149.65 | 146 |
| FCR (kg feed/kg of eggs produced) | 2.197 | 1.979 | 2.078 | 2.314 | 2.213 | 2.017 | 1.923 | 3.72 | 5.304 | 4.966 |
| Wastewater (kg) | 25 | 52.5 | 830 | 954.46 | 922.7 | 240 | 80 | 0 | 0 | 0 |
| $N_2O$ (kg/hen) | 0.0186 | 0.0156 | 0.0153 | 0.016 | 0.016 | 0.0194 | 0.0201 | 0.0166 | 0.0259 | 0.0265 |
| $NH_3$ (kg/hen) | 0.555 | 0.466 | 0.458 | 0.478 | 0.478 | 0.312 | 0.323 | 0.266 | 0.415 | 0.425 |
| $NO_3^-$ (kg/hen) | 0.723 | 0.607 | 0.596 | 0.623 | 0.623 | 0.766 | 0.794 | 0.655 | 1.020 | 0.861 |
| $CH_4$ (kg/hen) | 0.03 | 0.03 | 0.03 | 0.03 | 0.03 | 0.015 | 0.015 | 0.015 | 0.015 | 0.015 |
| $P_2O_5$ (kg) | 0.018 | 0.015 | 0.014 | 0.015 | 0.015 | 0.014 | 0.015 | 0.007 | 0.009 | 0.0001 |

**Table 3.** Feed composition expressed in percentage of total consumption.

| Ingredient | Rearing | Battery/Barn/Free Range | Organic 1 [1] | Organic 2 | Community Garden |
|---|---|---|---|---|---|
| Grounded maize | 10 | 8.38 | - | - | - |
| Grounded wheat | 57.84 | 49.03 | 47 | - | - |
| Grounded soy | 12.7 | 10.72 | - | - | - |
| Grounded triticale | 4.1 | 7.61 | - | 40 | - |
| Rapeseed pellets | 3.9 | 3.68 | - | - | - |
| Maize meal [2] | 0 | 3.09 | - | - | - |
| Sunflower meal | 4.5 | 3.77 | - | - | - |
| Animal fat | 0 | 1.05 | - | - | - |
| Calcium carbonate | 4.5 | 4.67 | - | - | - |
| Grass and worms | - | - | 23 | - | 25 |
| Grain mix (12.3% CP [3]) | - | - | - | - | 75 |
| Potato | - | - | 16 | - | - |
| Oats | - | - | - | 10 | - |
| Grain mix (20.9% CP) | - | - | - | 50 | - |

[1] The remaining 14% is left over from the farm: whey from goat cheese, and food waste from bed and breakfast that was not quantified by the farm. [2] Subproduct from bioethanol production. [3] Crude protein.

### 2.3. Life Cycle Impact Assessment

The impact assessment method Environmental Footprint 3.0 [24] was used to quantify impact categories. The categories assessed are acidification, climate change, ecotoxicity (freshwater), eutrophication (freshwater, marine and terrestrial), human toxicity (cancer and non-cancer), ionizing radiation (human health (HH)), land use, ozone depletion, particulate matter, photochemical ozone formation (POF, HH), resource use (fossils, mineral and metals), and water use. To clarify readings, when figures are shown using normalized and weighted results, the different categories of eutrophication, human toxicity, and resource use were added.

The model was divided into six subsystems: energy, feed, hatchery, rearing, laying, and transport. Feed includes feed from both rearing and laying, and energy includes heat and electricity used for all the processes. For results analysis, an average of all farms from a specific laying system was calculated.

### 2.4. Alternative Scenarios

Different scenarios were created to evaluate how key parameters influence impacts, and to what extent these could potentially be reduced.

Firstly, as feed was found to be the most impactful subsystem, Scenario 1 assesses the replacement of feed from battery, barn, and free-range by 70% of organic feed for the laying system, and 80% for rearing (corn, wheat and soybean). The quantity of feed was not changed.

Secondly, with a functional unit of one kilogram of shelled eggs, yield highly influences the results and organic farms evaluated in this study are only representative of small farms. Consequently, Scenario 2 evaluates the potential impact of organic farms with a yield corresponding to eggs produced in battery systems. The average yield for battery systems is 329 eggs/hen/year.

## 3. Results

### 3.1. Life Cycle Inventory of Resource Use

Using Environmental Footprint 3.0 for the resource use of fossils, for all laying systems, feed production contributes the most to resource depletion. The resting contribution is mainly allocated to electricity production. The detail of non-renewable energy resources share in feed and electricity production can be seen in Figure 1. For all laying systems, crude oil and lignite are responsible for the main impact of feed production and electricity production, respectively. The corresponding energy use is detailed in Table 4. It is higher

for organic and community garden systems, as more pullets are needed to produce one kilogram of eggs.

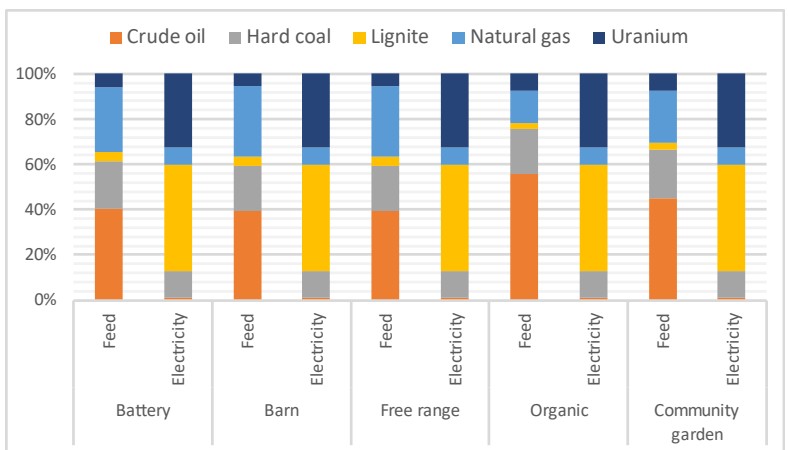

**Figure 1.** Relative contribution of non-renewable energy resources in feed and electricity production for each laying system for the production of one kilogram of eggs.

**Table 4.** Energy use for the production of one kilogram of eggs per laying system.

| Energy Resources (MJ) | Battery | Barn | Free Range | Organic | Community Garden |
|---|---|---|---|---|---|
| Non-renewable energy resources | 13.33 | 14.97 | 14.03 | 20.29 | 17.16 |
| Renewable energy resources | 44.18 | 48.19 | 42.55 | 111.15 | 70.54 |

Regarding the depletion of non-renewable materials, the main elements depleted for all laying systems are gold (approximately 90%), bromine, copper, iodine, lead, silver, and sulfur. More than 99% of this depletion is caused by feed production.

### 3.2. Impact Assessment of Laying Systems

The quantified results for the impact categories assessed and for each laying system can be found in Table 5. The detailed results for all impact categories and each farm are listed in the supplementary material Table S1.

**Table 5.** Results per impact categories for the production of one kilogram of eggs per laying system in the Czech Republic using Environmental Footprint 3.0.

| Impact Category | Battery | Barn | Free Range | Organic | Community Garden |
|---|---|---|---|---|---|
| Acidification (Mole of H$^+$ eq.) | 0.111 | 0.123 | 0.086 | 0.198 | 0.165 |
| Climate change (kg CO$_2$ eq.) | 2.46 | 3.45 | 3.21 | 3.46 | 3.48 |
| Ecotoxicity, freshwater (CTUe) | 62.87 | 60.25 | 52.35 | 50.5 | 95.1 |
| Eutrophication, freshwater (kg P eq.) | 0.000424 | 0.000477 | 0.00042 | 0.000944 | 0.00119 |
| Eutrophication, marine (kg N eq.) | 0.029 | 0.030 | 0.028 | 0.11 | 0.083 |
| Eutrophication, terrestrial (mole of N eq.) | 0.489 | 0.537 | 0.377 | 0.876 | 0.723 |
| Human toxicity, cancer (CTUh) | $2.31 \times 10^{-9}$ | $2.22 \times 10^{-9}$ | $1.96 \times 10^{-9}$ | $6.19 \times 10^{-9}$ | $4.29 \times 10^{-9}$ |
| Human toxicity, non-cancer (CTUh) | $1.4 \times 10^{-7}$ | $1.47 \times 10^{-7}$ | $1.3 \times 10^{-7}$ | $4.79 \times 10^{-7}$ | $1.51 \times 10^{-7}$ |
| Ionising radiation, HH (kBq U235 eq.) | 0.078 | 0.084 | 0.081 | 0.14 | 0.12 |
| Land use (Pt) | 264.28 | 349 | 320.5 | 368.12 | 562.49 |
| Ozone depletion (kg CFC-11 eq.) | $8.46 \times 10^{-8}$ | $8.34 \times 10^{-8}$ | $7.34 \times 10^{-8}$ | $1.38 \times 10^{-7}$ | $1.47 \times 10^{-7}$ |
| Particulate matter (disease incidences) | $8.67 \times 10^{-7}$ | $9.53 \times 10^{-7}$ | $6.89 \times 10^{-7}$ | $1.36 \times 10^{-6}$ | $1.26 \times 10^{-6}$ |
| POF, HH (kg NMVOC eq.) | 0.0051 | 0.0054 | 0.0049 | 0.0096 | 0.0087 |
| Resource use, fossils (MJ) | 13.33 | 14.95 | 14.05 | 20.3 | 17.16 |
| Resource use, mineral and metals (kg Sb eq.) | $3.56 \times 10^{-5}$ | $3.55 \times 10^{-5}$ | $3.13 \times 10^{-5}$ | $5.97 \times 10^{-5}$ | $6.95 \times 10^{-5}$ |
| Water use (m$^3$ world eq.) | 5.04 | 5.54 | 4.90 | 8.09 | 11.80 |

Figure 2 represents the contribution of subsystems to the overall impact for each laying system, and Figure 3 details impacts per category evaluated. For all laying systems, the feed represents approximately 50% of the total impact. The laying house makes up approximately 30% of this total, while hatchery, rearing, energy, and transport represent approximately 20%.

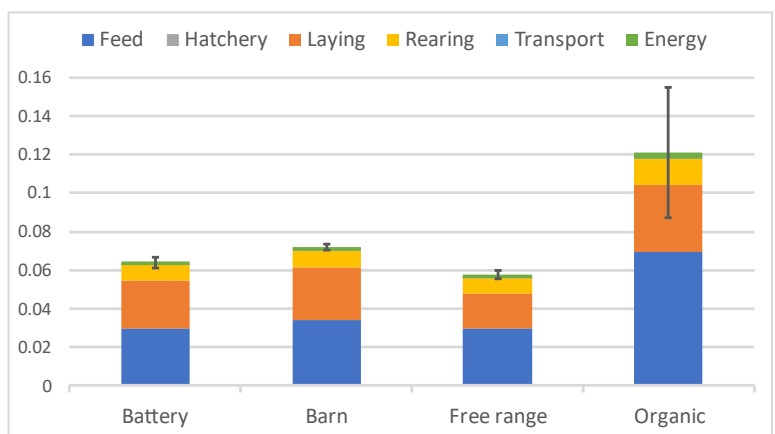

**Figure 2.** Normalized and weighted results for the different laying systems using Environmental Footprint 3.0 for one kilogram of eggs.

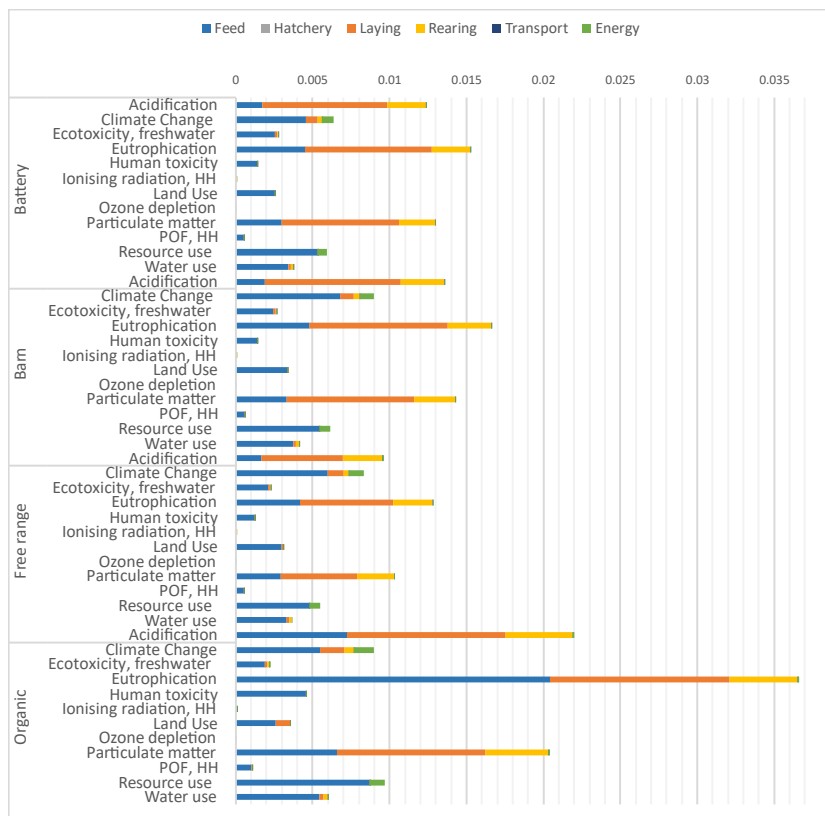

**Figure 3.** Normalized and weighted results for all impact categories for the different laying systems using Environmental Footprint 3.0 for one kilogram of eggs.

For one kilogram of eggs, the organic production has a higher environmental impact compared to others. The production of one kilogram of eggs requires more laying hens if the number of eggs that hens are laying is lower. This difference in egg laying affects feed

production, as more feed is needed if the yield is lower. This corresponds to the FCR, the quantity of feed required to produce one kilogram of eggs.

While battery, barn, and free-range had similar results, the two organic farms evaluated led to distinct outcomes. As shown in Figure 2, the organic farm with the lowest environmental impact is comparable to farms from other laying systems showing the highest environmental impacts. Overall, free-range eggs have the lowest environmental impact, mainly due to a lower FCR, but also due to manure management practices (explained in Section 3.2.4).

For each laying system, ionizing radiation, ozone depletion, and POF have only minor impacts in contrast to acidification, eutrophication, and particulate matter, as shown in Figure 3.

### 3.2.1. Feed

Feed is the most impactful subsystem for most of the impact categories, and is responsible for more than 90% of the impact in the categories of water use, land use, and resource use. The results are mainly linked to the FCR of each laying system. Consequently, the overall impact of organic feed is higher than conventional feed, besides the fact that organic laying hens consume more feed (about 150 g/head/day in contrast to less than 120 g/head/day for other laying systems).

The difference among impact categories such as acidification, particulate matter, human toxicity, and eutrophication between organic feed and conventional feed is due to a lower yield of organic feed. On the contrary, greenhouse gas emissions and land use are lower for organic feed (except for battery, due to the higher yield). Despite this contrast in yields, ecotoxic compounds emissions are lower for organic feed.

Regarding feed composition, the ingredients having the most impact are wheat followed by soybean and maize, due to the quantity consumed. We can also notice that impacts of rapeseed, sunflower meal, and animal fat are comparable to those of maize, while composing only 4%, 4%, and 1%, respectively, of the feed modeled. Likewise, and for ingredients included in the model, oats in the second organic farm, although representing only 10% of the diet, share the highest environmental impact.

### 3.2.2. Hatchery

Hatchery is the subsystem which has the least impact. The main emissions are related to heat and electricity use (see Section 3.2.6). Other minor impacts are due to land and water use.

### 3.2.3. Rearing

The impacts of rearing are linked to the excreted manure and heat used for rearing pullets (reported in the energy subsystem). Emissions from pullets are lower compared to laying hens due to the lower amount of feed eaten, and thus of nutrients excreted in manure. Pullets' manure results in the highest impacts in the categories of eutrophication, acidification, and particulate matter.

### 3.2.4. Laying

Laying is the second subsystem contributing the most to the total environmental impact, largely inducing acidifying, eutrophication, and particulate matter emissions. These emissions are determined by the proportion of crude protein and phosphorus in feed, which generates nitrogen and phosphorus excretions in manure. Here, we can note a strong variation in nitrogen excreted by hens in the two organic farms evaluated. Indeed, one of the farms uses a grain mix with a high content of crude protein (about 21%), which generates 2.6 g N/head/day, in comparison to 1.7 g N/head/day for the second organic farm, and around 2.0 g N/head/day for conventional hens. This reflects the diversity that small organic farms can have in terms of environmental impacts as feed differs largely between farms of this small scale—unlike industrial farms, where this parameter is more

controllable. Still, the FCR remains determinant and overall impact of organic laying hens is higher than conventional, since the FCR is much higher for this system.

However, it is interesting to observe that emissions of acidifying, eutrophication, and particulate matter compounds are lower for free-range hens than battery, even with a similar FCR. This is caused by the amount of manure deposited on grassland, which does not need to be managed, reducing ammonia emissions. On the other hand, greenhouse gas emissions are higher for organic and free-range laying hens due to higher nitrous oxide emissions, its emission factor being higher for nitrogen deposited on grassland than for nitrogen managed or applied to soil.

The lower impact of organic feed on land use is balanced by a slightly greater land use of organic laying houses.

### 3.2.5. Transport

After hatchery, the transport included in this study has the lowest impact among sub-systems. Greenhouse gas emissions followed by resource use share the main contribution.

### 3.2.6. Energy

Electricity and heat consumption contribute by approximately 10% to climate change, and have only minor impacts in other categories. Heat is used for hatchery and rearing, while electricity is used in most processes and has a higher impact than heat. For both organic laying systems, electricity and heat are used only for rearing pullets and not in the laying house, but the impact of energy on climate change is still higher for organic eggs, due to the much higher number of hens needed to produce one kilogram of eggs.

### 3.3. Impact Assessment of the Community Garden

In Figure 4, the environmental impacts of eggs produced in the community garden are compared with free-range and organic eggs. The comparison with free-range eggs is relevant, as eggs are produced in the same laying system, at different scales, while the comparison with organic eggs is more alike in terms of the scale of the farms. The total impact of eggs from the community garden is lower than the impact of organic eggs but higher than free-range eggs. Reflecting previous results, feed is responsible for half of the total impact, while laying is responsible for most of the other half.

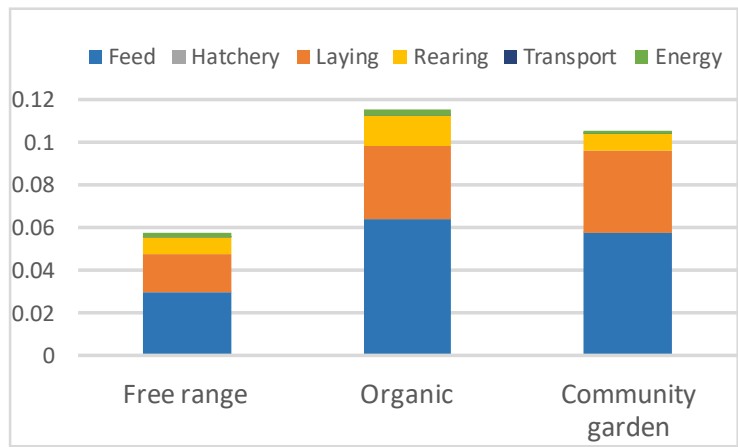

**Figure 4.** Normalized and weighted results for eggs produced in a community garden compared with free-range and organic eggs, using Environmental Footprint 3.0 for one kilogram of eggs.

Acidifying, eutrophication, and particulate matter emissions of the community garden are a slightly lower to those of organic eggs due to the use of conventional feed, as can be seen in Figure 5. Interestingly, the FCR of the community garden is higher than for organic eggs, but the impact of feed is lower. Unlike the previous impact assessment where the yield was mostly influencing impacts, here, despite the low yield of the eggs from

the community garden, the overall impact is lower than organic eggs due to the use of conventional feed, as well as a lower crude protein content in the diet.

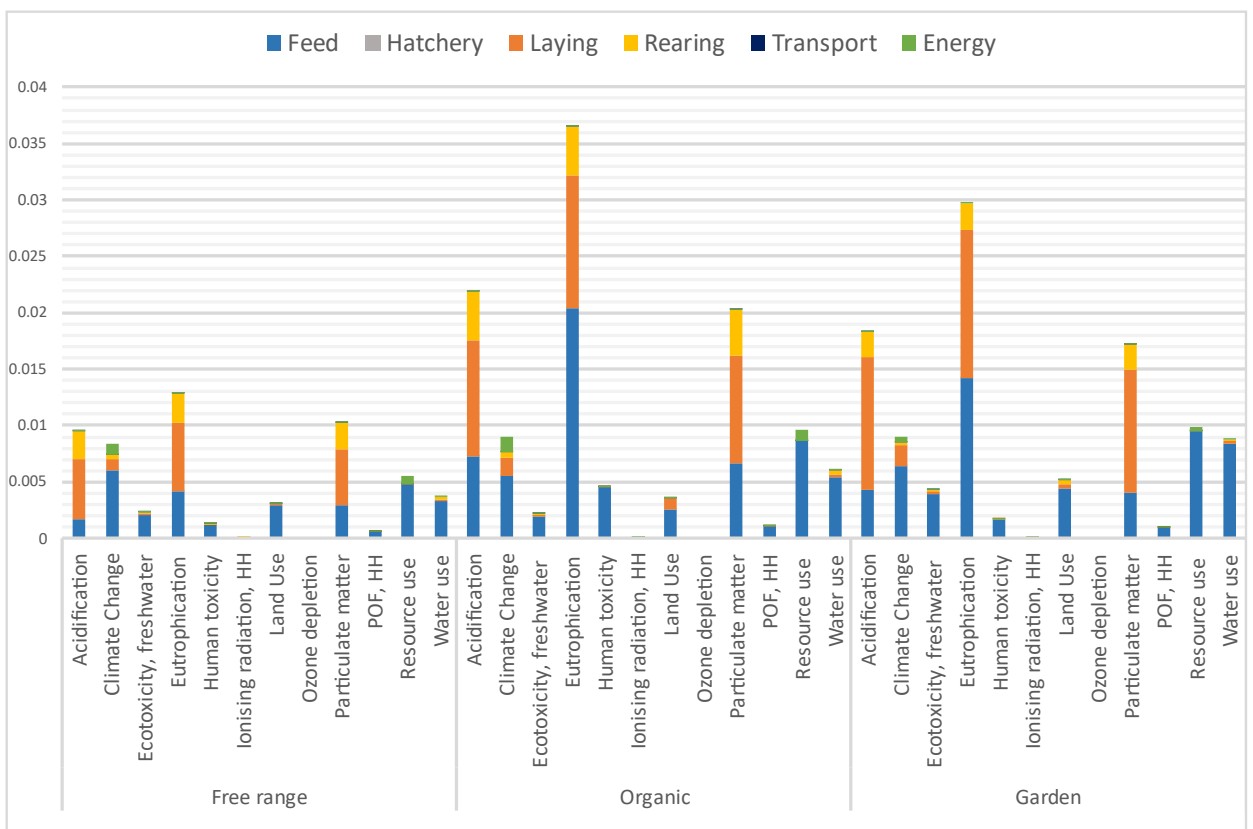

**Figure 5.** Normalized and weighted results for eggs produced in a community garden compared with free-range and organic eggs, using Environmental Footprint 3.0 for one kilogram of eggs per impact categories.

### *3.4. Alternative Scenarios*

#### 3.4.1. Scenario 1

In Figure 6a, the baseline is set at 100%, and represents the results from previous impact assessment in Section 3.2. It shows a potential for considerable reductions of impacts in the categories of ecotoxicity and land use. If organic feed replaces conventional feed in battery, barn, and free-range systems, impacts in these categories could be reduced by more than half. Greenhouse gas emissions and resource use are reduced by up to 40% and 20% respectively. However, the most impactful categories are increased, as well as human toxicity.

With all laying systems using organic feed, free-range is still the most sustainable option in all impact categories, as can be seen in Figure 6b.

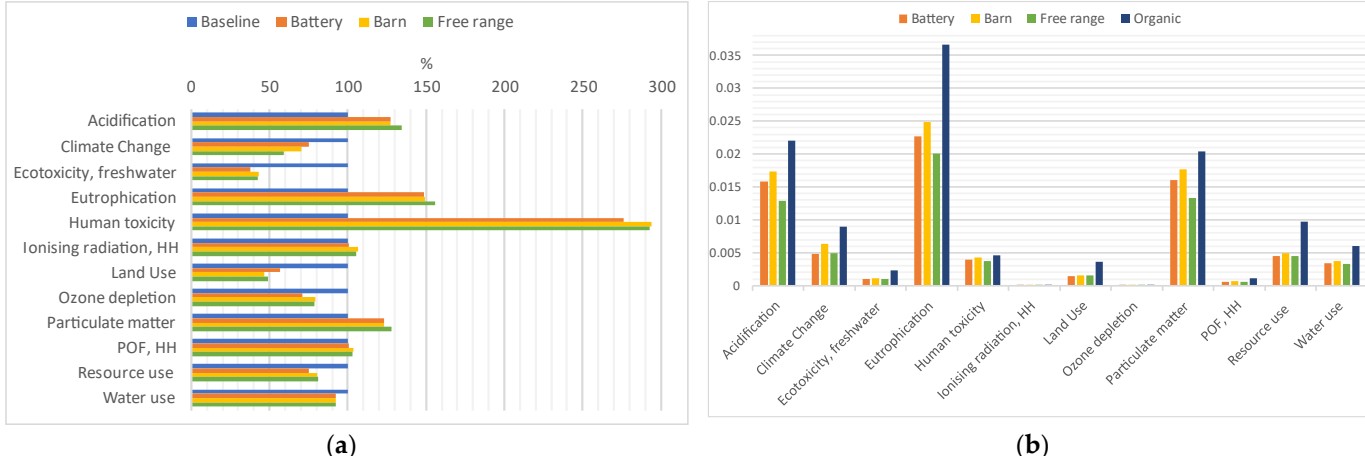

(**a**)                                                                 (**b**)

**Figure 6.** Potential reduction in impact categories if conventional feed is replaced by organic feed using Environmental Footprint 3.0, for one kilogram of eggs: (**a**) for battery, barn, and free-range compared to the baseline; (**b**) normalized and weighted results for battery, barn, and free-range compared to organic eggs.

### 3.4.2. Scenario 2

Figure 7a shows that if the yield of organic eggs was higher, all impact categories would be reduced by 50%. In this scenario, which can be seen as industrial organic eggs, the total environmental impact of organic eggs is comparable to other laying systems, as is shown in Figure 7b. The impact of organic feed remains higher than conventional feed, but emissions occurring during laying and rearing are lower. The free-range system remains more environmentally friendly, but the lowest range for organic eggs (which represents the organic farm with a lower crude protein content) has the lowest environmental impact.

Figure 8 details the results for each impact category. In this scenario, we see the potential mitigation in all impact categories, except acidification and eutrophication, which once more depend on protein content of the feed.

These results represent potential impacts for industrial organic eggs.

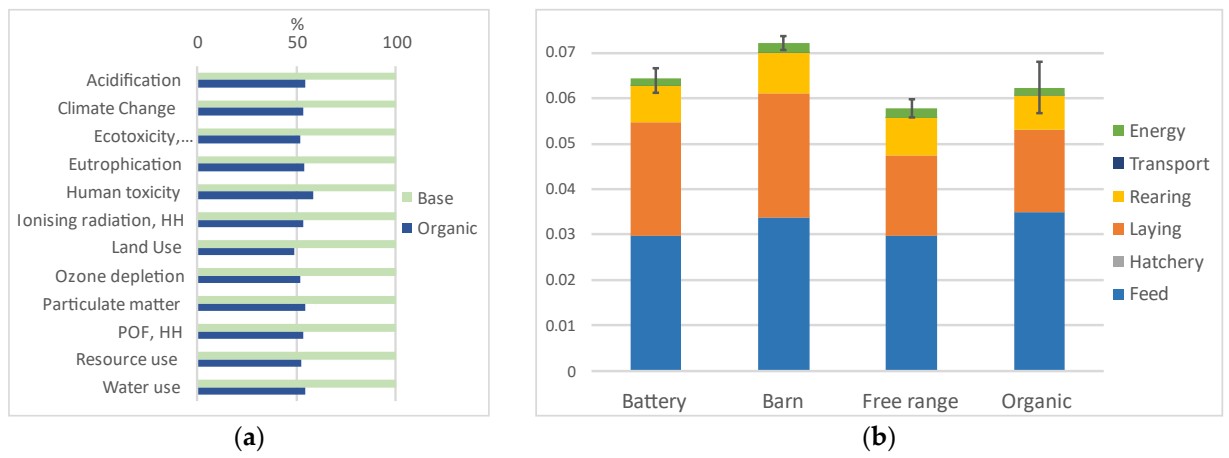

(**a**)                                                                 (**b**)

**Figure 7.** Changes in impact categories for organic eggs with the same yield as eggs produced in a battery system using Environmental Footprint 3.0, for one kilogram of eggs: (**a**) compared to the baseline; (**b**) normalized and weighted total results compared to battery, barn, and free-range eggs.

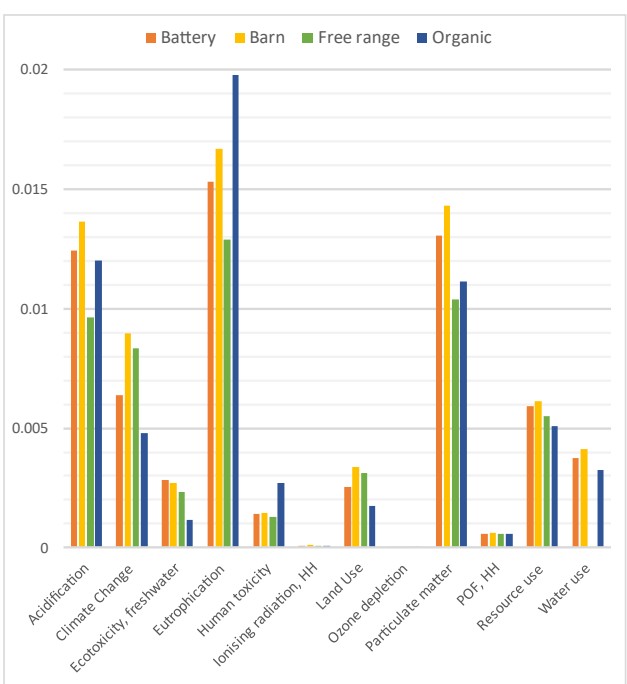

**Figure 8.** Normalized and weighted results for each impact category and laying systems for organic eggs that have a yield similar to battery eggs, using Environmental Footprint 3.0 for one kilogram of eggs.

## 4. Discussion

Comparisons with other LCA studies should be conducted with caution due to the different assumptions made and methodology used. The main hotspot of environmental impacts is feed for all LCA studies on egg production. More generally, the comparison of the climate change category reported in Table 6 is dependent on the FCR, which varies across countries and systems. For all the countries, battery eggs had lower greenhouse gas emissions compared to organic eggs, except for Pelletier et al. [11], which reported the lowest emission for organic eggs, due to similar FCR between laying systems. The impact of Italian organic eggs is lower compared to the Czech ones, more than two times lower, though the FCR is also two times lower. This lower FCR for Italian organic eggs is a consequence of two factors: the lower daily intake of feed, and a much higher number of eggs laid (328.5 eggs/hen/day in comparison to an average of 188.83 for Czech organic eggs). This shows a potential for decreasing impacts if the FCR of organic eggs was reduced in the Czech Republic, as presented in Scenario 2 (1.84 kg $CO_2$ eq.). Our results are alike the results obtained by Leinonen et al. [10] in the United Kingdom. They also reported lower feed intake for laying hens from battery systems, as well as higher yield compared to organic laying hens, although higher than those of the Czech Republic. The FCR is lower for English organic eggs, but interestingly, greenhouse gas emissions are the same in both countries. This is because, for one kilogram of eggs, the production of organic feed in the Czech Republic results in lower greenhouse gas emissions than in the United Kingdom.

The most determinant factors for environmental impacts of eggs are the FCR and feed composition. Using organic feed instead of conventional feed could potentially reduce environmental impacts. However, a way to cope with acidification, eutrophication, and particulate matter pollution should be found. Meier et al. [25], as well as Avadí et al. [26] more recently, reported that the methodology used is not adapted to organic systems, casting doubts about the fact that higher burdens of organic systems per product unit are caused by lower yields. In particular, the assumptions made for nitrogen emissions calculations are the same for conventional and organic systems, even though the mode of action of organic systems is different. This shows that there is still progress to be made in

the assessment of organic systems. Similarly, the integration of crop rotation, soil quality, biodiversity, or else multifunctionality to LCA would clarify the results.

**Table 6.** Climate change potential in kg $CO_2$ eq. and feed conversion ratio for one kilogram of eggs; a comparison with other LCA studies.

| Study | Country | | Battery | Barn | Free Range | Organic |
|---|---|---|---|---|---|---|
| Pelletier et al. | Canada | FCR | 2.2 | 2.1 | 2.2 | 2.0 |
| | | Kg $CO_2$ eq | 2.31 | 2.4 | 2.4 | 1.37 |
| Rocío et al. | Spain | FCR | 2.8 | - | - | - |
| | | Kg $CO_2$ eq | 3.4 | - | - | - |
| Constantini et al. | Italy | FCR | | | | 2.49 |
| | | Kg $CO_2$ eq | - | - | - | 1.46 |
| Leinonen et al. | United Kingdom | FCR | 2.15 | 2.4 | 2.55 | 2.69 |
| | | Kg $CO_2$ eq | 2.92 | 3.45 | 3.38 | 3.42 |
| Dekker et al. | Netherlands | FCR | 1.99 | 2.28 | 2.33 | 2.59 |
| | | Kg $CO_2$ eq | 2.24 | 2.67 | 2.74 | 2.55 |
| Our study | Czech Republic | FCR | 2.08 | 2.26 | 1.97 | 4.51 |
| | | Kg $CO_2$ eq | 2.46 | 3.45 | 3.41 | 3.46 |

The nitrogen in feed composition also influences manure emissions, which is the second most impactful factor. These emissions could be reduced especially by adapting the quantity of crude protein to the need of hens. This is especially true for small farms. Note that phosphorus content was found to have only a minor impact on eutrophying emissions (about 1% of the overall impact). As seen with free-range laying hens, the emissions related to manure management could be mitigated by having a laying house with a possibility for hens to go outside. This would reduce ammonia emissions, ammonia being a strong acidifying and eutrophying compound.

Unlike feed and laying, the subsystems of hatchery, rearing, energy, and transport were only a minor contribution to the overall impact. However, transport of feed was not included in this study, as it is highly variable. Considering how feed is transported could improve the precision of the results.

Some other assumptions made in this study could have an impact on the results. For example, for organic eggs, we considered that farmers were buying pullets, but this is not the case for all organic farms, as some of them are letting hens reproduce; this may decrease environmental impacts, as energy inputs would be removed and number of feed ingredients reduced.

To sum up, in this study, we evaluated the environmental impacts of one kilogram of eggs in the Czech Republic. Free-range eggs were found to be the most environmentally friendly option, thanks to a lower FCR combined with manure management practices. However, it should be noted that laying hen systems present ethical controversies.

In the future, it would be interesting to choose a different functional unit. A mass-based functional unit, even if relevant for industries, might not be the case for consumers, as it favors quantity over quality. Another functional unit, such as the amount of protein or another significant nutrient, could be used. In addition to their influence on environmental impacts, laying hen systems might influence nutritional aspects of eggs [27]. Similarly, the use of drugs, which was not included in this research, might differ between egg-laying systems. While some might use them as prevention, others use them only when animals are sick. However, in 2022, the EU banned the preventive use of antibiotics in farming [28]. According to this report, another market is growing: the use of slower-growing genotypes, especially for free-range and organic systems. The use of this kind of breed in the Czech Republic could affect the FCR and thus the environmental impacts.

## 5. Conclusions

To the best of our knowledge, we are the first ones to use LCA to compare the four egg laying systems in the Czech Republic. In this cradle-to-gate analysis, we found that FCR, feed composition, and manure management were the most important factors affecting the total environmental impacts of eggs, irrespective of the laying system. On the contrary, transport and energy were found to have only minor impacts. Greenhouse gas emissions and ecotoxicity were mostly caused by feed, and could be reduced by using organic feed over conventional feed, while emissions from manure could be mitigated by adapting the crude protein content of diets or switching to a laying system with outdoor access. Furthermore, organic eggs have more significant environmental impacts than conventionally produced eggs, though more specific modeling for nitrogen emissions adapted to organic systems should be applied in order to obtain more accurate results.

As there is drive for the EU egg industry to become cage-free, the data of our study may be used while issuing recommendations for farms with enriched cages that will undergo transformation into another laying system in the near future.

**Supplementary Materials:** The following supporting information can be downloaded at: https://www.mdpi.com/article/10.3390/agriculture12030355/s1, Table S1: Total results for each impact categories and farms for the production of one kilogram of eggs, using Environmental Footprint 3.0.

**Author Contributions:** Conceptualization, A.G. and A.H.-V.; Data curation, A.G. and A.H.-V.; Formal analysis, A.G.; Funding acquisition, A.G. and A.H.-V.; Investigation, A.G. and A.H.-V.; Methodology, A.G. and A.H.-V.; Project administration, A.H.-V.; Resources, V.K.; Software, A.G.; Supervision, V.K.; Validation, A.H.-V. and V.K.; Visualization, A.G. and A.H.-V.; Writing—original draft, A.G. and A.H.-V.; Writing—review & editing, A.G. and A.H.-V. All authors have read and agreed to the published version of the manuscript.

**Funding:** This research was funded by internal grants of UCT Prague, grant numbers A2_FTOP_2021_017 and A1_FTOP_2021_003.

**Institutional Review Board Statement:** Not applicable.

**Informed Consent Statement:** Not applicable.

**Data Availability Statement:** Publicly available datasets were analyzed in this study. This data can be found here: https://portal.cenia.cz/eiasea/view/eia100_cr (accessed on 29 November 2021).

**Acknowledgments:** We would like to acknowledge the farms who agreed to help us by sharing some information and data on their activity. We would also like to thank our colleagues for their support.

**Conflicts of Interest:** The authors declare no conflict of interest. The funders had no role in the design of the study; in the collection, analyses, or interpretation of data; in the writing of the manuscript, or in the decision to publish the results.

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
