# Peer review of "Environmental Impacts of Egg Production from a Life Cycle Perspective"

_agriculture, doi:10.3390/agriculture12030355_

Round 1

Reviewer 1 Report

Line 109, System boundaries for egg production, seems not clear enough

Line 441, the conclusion part , too long, make it simple

Line 481, make the reference format consistant

Some figures lack the description of  y-axis, make them complement

Author Response

Dear Reviewer, 

Thank you for all your comments and suggestions. The following changes were made:

-line 109, new title of Scheme I is "System boundaries for the cradle-to-gate LCA of Czech egg production"

-line 441, the conclusion was shorten

-line 481, the references were changed by downloading endnote reference format "MDPI"

For normalized results, the results in y-axis don't have any units and are meant only for comparison of impact categories. However, the y-axis was added for all figures.

Thank you again and please let us know if something is still unclear. 

Reviewer 2 Report

Dear Authors, I found your work very interesting and important in terms of content. I feel a little scarcity and I had some assumptions to FCR presentation but you use estimated data so I leave this part of my comments. I think this work is a good reference point for those who will rate some real parameters not from database. But of course this does not belittle the value of your innovative work. The amount of work is noticeable and I have only few suggestions which can help a little to improve it for better clarity. Perhaps in the end you can more definite summary, I think about sometimes our to simply evaluation and comparison the  intensive and extensive systems with regard to environmental protection.

so first for visual quality:

1) Fig.2 in my opinion you should decide which options you prefer and leave only one a or b because for interpretation /conclusion this two figures represent the same

2) Fig.3 isn't citied in the text, we have summarizing sentence (l.254-255), correct it. I think that it will be better to change this presentation into vertical (like Fig.6a)

3) Tab. 4 - Energy use for the production of one kilogram of eggs per laying system,  remove from this table detailed non-renewable energy sources (you repeat the same data and it could be confusing), leave only two lines with non-renewable and renewable energy source

4) I don't like title of Tab.5. Perhaps it could be: the calculated impact categories for each ...., and in legend you can introduce the explanation to this results

5) l.: 268-273; I would be careful with such summing, especially when you describe organic farms, apart from solid feed birds had access to other nutritive components like grass, worms, etc.,  and it is impossible to assess their content /part in total daily ration. So in the end it is very difficult to give  the final interpretation. Add sth more in discuss about it.

6) l. 323 " As discussed"..., in this part you present Results not Discuss  

7) l. 332-3 - is  there a full sentence?, it is difficult to understand it, perhaps change it a little

Author Response

Dear reviewer, 

Thank you for all your inputs and suggestion for improvement. Here you can see a detail of the changes made:

-fig 2: fig.a was kept as it also provide detailed information to compare between systems. Line 243: figure 2a was replaced by figure 2.

-fig3: it is cited at line 234, but "as shown in the Figure 3" was added line 255. The fig3 was changed to vertical. Please confirm if it should be kept this way. If yes, should we also change figure 5 in a vertical way?

-table 4: two lines were kept (non-renewable/renawable energy), the rest was deleted

-table 5: the title was changed to "Results per impact categories for the production of one kilogram of eggs per laying system in the Czech Republic using Environmental Footprint 3.0."

-line 273: we detailed that it is only valid for ingredients included in the model

-line 324: it was changed to "Like previous results.."

-line 333: the sentence was detailed: "Acidifying, eutrophying and particulate matter emissions of the community garden are a bit lower to those of organic eggs due to the use of conventional feed, as can be seen in the Figure 5. "

Thank you again for your inputs and let us know if something is still unclear. 

Reviewer 3 Report

The paper entitled "Environmental Impacts of Egg Production from Life Cycle Perspective" is very interesting and provides an excellent comparison between the types of chicken rearing and feeding systems.
The data obtained are very interesting, the simulations show important directions for making decisions that generate less environmental impact.

Author Response

Dear Reviewer,

Thank you very much for your comments.